# A Novel Automated Curriculum Strategy to Solve Hard Sokoban Planning Instances

**Dieqiao Feng**
Department of Computer Science
Cornell University
Ithaca, NY 14850
dqfeng@cs.cornell.edu

**Carla P. Gomes**
Department of Computer Science
Cornell University
Ithaca, NY 14850
gomes@cs.cornell.edu

**Bart Selman**
Department of Computer Science
Cornell University
Ithaca, NY 14850
selman@cs.cornell.edu

## Abstract

In recent years, we have witnessed tremendous progress in deep reinforcement learning (RL) for tasks such as Go, Chess, video games, and robot control. Nevertheless, other combinatorial domains, such as AI planning, still pose considerable challenges for RL approaches. The key difficulty in those domains is that a positive reward signal becomes *exponentially rare* as the minimal solution length increases. So, an RL approach loses its training signal. There has been promising recent progress by using a curriculum-driven learning approach that is designed to solve a single hard instance. We present a novel *automated* curriculum approach that dynamically selects from a pool of unlabeled training instances of varying task complexity guided by our *difficulty quantum momentum* strategy. We show how the smoothness of the task hardness impacts the final learning results. In particular, as the size of the instance pool increases, the "hardness gap" decreases, which facilitates a smoother automated curriculum based learning process. Our automated curriculum approach dramatically improves upon the previous approaches. We show our results on Sokoban, which is a traditional PSPACE-complete planning problem and presents a great challenge even for specialized solvers. Our RL agent can solve hard instances that are far out of reach for any previous state-of-the-art Sokoban solver. In particular, our approach can uncover plans that require hundreds of steps, while the best previous search methods would take many years of computing time to solve such instances. In addition, we show that we can further boost the RL performance with an intricate coupling of our automated curriculum approach with a curiosity-driven search strategy and a graph neural net representation.

## 1 Introduction

Planning is an area in core artificial intelligence (AI), which emerged in the early days of AI as part of research on robotics. An AI planning problem consists of a specification of an initial state, a goal state, and a set of operators that specifies how one can move from one state to the next. In robotics, a planner can be used to synthesize a plan, i.e., a sequence of robot actions from an initial state to a desired goal state. The generality of the planning formalism captures a surprisingly wide range of tasks, including task scheduling, program synthesis, and general theorem proving (actions are

inference steps). The computational complexity of general AI planning is at least PSPACE-complete [4, 6]. There are now dozens of AI planners, many of these compete in the regular ICAPS Planning competition [19]. In this paper, we consider how deep reinforcement learning (RL) can boost the performance of plan search by *automatically* uncovering domain structure to guide the search process.

A core difficulty for RL for AI planning is the extreme sparsity of the reward function. For instances whose shortest plans consist of hundreds of steps, the learning agent either gets a positive reward by correctly finding the whole chain of steps, or no reward otherwise. A random strategy cannot "accidentally" encounter a valid chain. In [11], we proposed an approach to addressing this issue by introducing a *curriculum-based* training strategy [2] for AI planning. The curriculum starts with training on a set of quite basic planning sub-tasks and proceeds with training on increasingly complex set of sub-tasks until enough domain knowledge is acquired by the RL framework to solve the original planning task. We showed how this strategy can solve several surprisingly challenging instances from the Sokoban planning domain.

Herein we introduce a novel *automated dynamic curriculum strategy,* which is more general and significantly extends the curriculum approach of [11] and allows for solving a broad set of previously unsolved planning problem instances. Our approach starts with a broad pool of sub-tasks of the target problem to be solved. In contrast to [11], our approach does require the manual identification of groups of increasingly hard sub-tasks. All sub-tasks are "unlabeled," i.e., no solution plans are provided and many may even be unsolvable. We introduce a novel multi-armed bandit strategy that automatically selects batches of sub-tasks to feed to our planning system or RL agent, which we refer to as *difficulty quantum momentum* strategy, using the current deep net policy function and a Monte Carlo Tree Search based search strategy. The system selects tasks that can be solved and uses those instances to subsequently update the policy network. By repeating these steps, the policy network becomes increasingly effective and starts solving increasingly hard sub-tasks until enough domain knowledge is captured and the original planning task can be solved. As we will demonstrate, the difficulty quantum momentum multi-armed bandit strategy allows the system to focus on sub-tasks that lie on the boundary of being solvable by the planning agent. In effect, the system dynamically uncovers the most useful sub-tasks to train on. Intuitively, these instances fill the "complexity gap" between the current knowledge in the policy network and the next level of problem difficulty. The whole framework proceeds in an unsupervised manner — little domain knowledge of the background task is needed and the bandit is being guided by a simple but surprisingly effective rule.

As in [11], we use Sokoban as our background domain due to its extreme difficulty for AI planners [16]. See Figure 1 for an example Sokoban problem. We have a 2-D grid setup, in which, given equal number of boxes and goal squares, the player needs to push all boxes to goal squares without crossing walls or pushing boxes into walls. The player can only move upward, downward, leftward and rightward. Sokoban is a challenging combinatorial problem for AI planning despite its simple conceptual rules. The problem was proven to be PSPACE-complete [7] and a regular size board (around $15 \times 15$) can require hundreds of pushes. Another reason for its difficulty is that many pushes are irreversible. That is, with a few wrong pushes, the board can become a dead-end state, from which no valid plan leading to a goal state exits. Modern specialized Sokoban solvers are all based on combinatorial search augmented with highly sophisticated handcrafted pruning rules and various dead-end detection criteria to avoid spending time in search space with no solution.

**Preview of Main Results** We evaluate our approach using a large Sokoban repository [23]. This repository contains 3362 Sokoban problems, including 225 instances that have not been solved with any state-of-the-art search based methods. [11] focused on solving single hard instances and showed how several such instances could be solved using a handcrafted portfolio strategy. Our focus here is on the full subset of unsolved instances and our automated dynamic curriculum strategy.

Table 3 provides a summary of our overall results. Our baseline strategy (BL) uses a convolutional network to capture the policy and samples uniformly from the sub-tasks. Our baseline can solve **30** of the **225** unsolved instances (**13%**), using 12 hours per instance, including training time. Adding curiosity rewards (CR) to the search component and using a graph neural net (GN), we can solve 72 instances (**32%**). Then, adding the multi-armed bandit dynamic portfolio strategy, enables us to solve 115 cases (**51%**), and, training on a pool of all open problem instances and their (unsolved) sub-cases together, lets us solve 146 instances (**65%**). Finally, we also added the remaining 3137 instances from the repository to our training pool. These are solvable by existing search-based solvers but we do not use those solutions. With these extra "practice problems," our automated curriculum deep

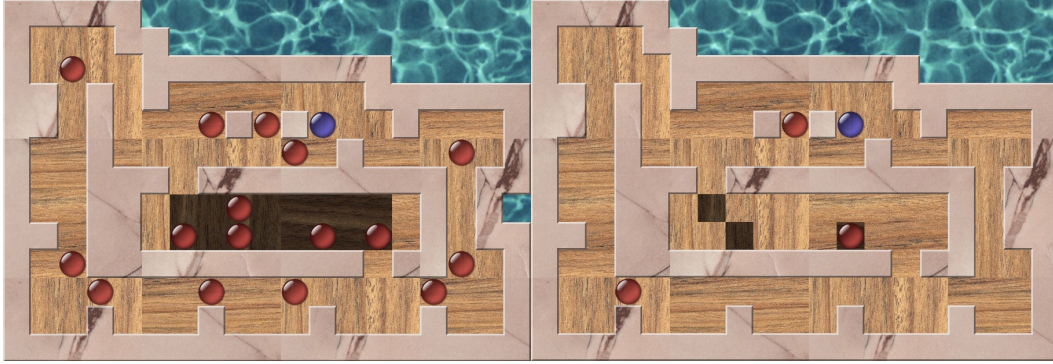

Figure 1: **Sokoban. Left panel**: An example of a Sokoban instance [11]. The blue circle represents the player, red circles represent boxes and dark cells are goal squares. Walls are represented by light colored cells. The player has to push all boxes to goal squares without going through walls. The player can only move upward, downward, leftward and rightward and push a box into a neighbor empty square and when placed in an empty square next to a box. **Right panel:** a subcase with 2 boxes and goal squares. To build a subcase, we first randomly pick the number of box/goal cells of the subcases, and then randomly select box/goal locations as a subset of initial box/goal locations.

RL planning approach can solve **179** out of 225 unsolved problems (**80%**). Many of the solutions require plans with several hundreds of steps. Moreover, these instances are now solved with a *single* deep policy network augmented with Monte Carlo tree search (MCTS). This suggests that the deep network *successfully captures a significant amount of domain knowledge about the planning domain.* This knowledge, augmented with a limited amount of combinatorial search (UCT mechanism), can solve a rich set of very hard Sokoban planning problems.

## 2 Related Work

As noted above, AI planning is a hard combinatorial task, at least PSPACE-complete. The recent remarkable success of deep RL on Go and Chess [22], which are also discrete combinatorial tasks, raises the prospect of using deep RL for AI planning. Key to the success of deep RL in multi-player combinatorial games is the use of *self-play* through which the RL agent obtains a useful learning signal and can gradually improve. In fact, the self-play strategy in a game setting, where each side is playing at the same strength, provides a natural training curriculum with a continually useful training signal (both wins and losses), enabling the learning system to improve incrementally. It is not clear how to obtain such a learning signal in a single-player setting such as AI planning. Starting directly on the unsolved hard planning task does not lead to any positive reinforcement signal because that would require MCTS to solve the instance (note that the initial deep policy network is random). However, as we will see, a training curriculum build up from simpler (unsolved) subcases of the initial planning problem can be used to bootstrap the RL process. We first introduced this idea in [11], using a handcrafted curriculum strategy. Our *automated dynamic curriculum strategy,* combined with several enhancements, substantially expands and outperforms the approach of [11].

Elman [10] first proposed that using a curriculum could improve the training speed of neural networks. The idea of curriculum prevailed in the deep learning community [2] due to the increasing complexity of tasks being considered. Graves et al. [12] used an automated strategy to select labeled training data points to accelerate training of neural networks in supervised learning. However, in the planning domain, ground truth plans usually are not available; moreover, the main focus of AI planning is on solving previously unsolved problems in reasonable time limit instead of accelerating solving easier problems. The curriculum strategy used in this paper is to push forward the border of feasible planning instances. Indeed, we will show that we can solve a large set of Sokoban instances whose combinatorial complexity exceeds the capability of state-of-the-art specialized Sokoban solvers.

Reward shaping is another approach to overcome the sparse reward issue. Instead of assigning positive reward only if the agent achieves the goal and non-positive reward otherwise, Ng et al. [17] used the idea of reward shaping to manually design extra positive rewards along the plan when the agent achieves some sub-goals. The strategy however requires domain-specific knowledge to provide

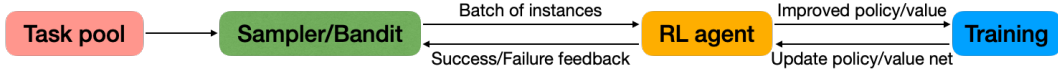

Figure 2: **The workflow of our automated curriculum framework.** In each iteration, the sampler/bandit picks a batch of task instances from the pool and the RL agent, which is based on Monte-Carlo tree search augmented with policy/value predictions, attempts to solve these instances. The success/failure status of each instance is sent back to the sampler/bandit to adjust its weights. Each successful attempt not only generates a valid solution but also improves policy/value data for the trainer to train the deep network of the agent.

---

**Algorithm 1:** Automated Curriculum Learning framework overview

---

**Input:** A Sokoban instance $\mathcal{I}$, solution length limit $L$, number of iterations $T$;
Create sub-instances from $\mathcal{I}$ to form a task pool;
**for** $t = 1, ..., T$ **do**
    Use uniform sampling or difficulty quantum momentum bandit to select a batch $B$ from the task pool;
    **for** $s \in B$ **do**
        **for** $i = 1, ..., L$ **do**
            Use MCTS to select the best move $a$ for $s$;
            $s = \text{next\_state}(s, a)$;
            **if** *s is a goal state* **then**
                Generate data for training the policy deep network of the RL agent;
                Send "success" feedback to the sampler/bandit;
                Break;
            **end**
        **end**
        **if** *Solution not found* **then**
            Send "failure" feedback to the sampler/bandit;
        **end**
    **end**
    Train the policy deep neural network of the RL agent;
**end**

---

fruitful rewards to make hard learning tasks feasible. In contrast to extrinsic rewards, intrinsic rewards such as curiosity [18, 9] can also help in exploring the search space more efficiently. The intrinsic rewards usually do not require domain specific knowledge, and the agent can achieve high scores on some Atari games purely guided by curiosity. However, the combinatorics of general AI planning is much more complex.

As our test domain, we use Sokoban, which is an notoriously hard AI planning domain. Deep neural networks have been used to tackle Sokoban, besides [11] mentioned above. Weber et al. [24] used an imagination component to help the reinforcement learning agent, and Groshev et al. [13] learned from Sokoban solutions and apply imitation learning to generalize to new instances. However, none of these approaches can perform close to modern specialized Sokoban solvers, such as Sokolution [8], which is based on backtrack search and incorporates a large variety of sophisticated domain specific heuristics.

## 3 Formal Framework

An overview of the workflow of our ***automated curriculum framework*** is depicted in Figure 2. We are interested in solving hard Sokoban instances, without labeled (solved) instance training data. Figure 1 shows an example of a Sokoban instance. To solve a hard Sokoban instance, our method first creates a pool of sub-instances, from the input instance(s) (as described in subsection 3.1), which is followed by multiple iterations of interactions between the multi-armed bandit and the RL agent as well as interactions between the RL agent and its policy deep neural network. In subsection 3.2, we describe how the multi-armed bandit generates batches of instances for the RL agent and

updates the sampling weight according to the feedback from the RL agent. In subsection 3.3 we provide further details on the RL model and the way we train it. In subsection 3.4, we describe curiosity-driven rewards and a graph neural network — two components that further improve our automated curriculum framework. A formal algorithm description can be found in Algorithm 1.

## 3.1 Sub-instance Creation

To set up a curriculum to solve a hard instance, extra (unlabeled) data with smoothly increasing difficulty is required. Unfortunately, in general, planning tasks do not have auxiliary data to support solving hard instances. To build a pool of extra training data we generate sub-instances from the original instance. Figure 1 shows a Sokoban instance (left panel) and one of its sub-instances (right panel). A sub-instance is generated by selecting a subset (of size $k$) of the initial boxes on their starting squares and selecting $k$ goal locations for those boxes.

This strategy has several advantages: (1) the number of data points we can sample grows exponentially as the number of boxes increases, which significantly facilitates learning; (2) The created subcases share common structure information with the original instance so the knowledge learned from the subcases generalizes better to the original one; (3) subcases with different number of boxes/goals naturally build a curriculum of instances of increasing difficulty. This enables a curriculum setup that trains on subcases with fewer boxes first and gradually moves to subcases with more boxes.

## 3.2 Curriculum Learning Setup

A key contribution of our work is to provide an **automated** strategy for the tuning of the difficulty and the selection of sub-instances for the training, which contrasts to the static, manually driven order used by [11]. In our automated curriculum framework, given a pool of Sokoban instances, a sampler/multi-armed bandit generates batches for the RL agent to solve. Ideally, we want to develop a curriculum of increasing difficulty by generating easier batches first and harder ones later. As shown in [11], the number of boxes of subcases is a good difficulty indicator and the static curriculum based on the increasing number of boxes shows good results. However, such a natural difficulty ordering won't exist in many other planning domains. Moreover, we can do better than a handcrafted approach. To build a more generalizable curriculum strategy, we sample training batches for the RL agent using a new multi-armed bandit strategy that we refer to as *difficulty quantum momentum*. This strategy selects instances at the edge of solvability. Other selection strategies often select instances that are too hard or too easy for the agent to solve. Neither provides a useful learning signal. Intuitively, our *difficulty quantum momentum* strategy prefers to select a batch whose difficulty just lies on the edge of the agent's capability, thus providing a significant learning signal to the RL agent. Our experiments will show the effectiveness of this approach.

**Difficulty quantum momentum strategy:** Unlike uniform sampling, the bandit attempts to learn the weight of each instance from the feedback of the RL agent. Assume the size of pool is $N$, for each task $T_i$ we maintain a scalar $h_i$ indicating the failure/success (0/1) history of the task and initialize $h_i$ to 0 for $i = 1, ..., N$ before learning. Once the agent tries $T_i$, we define the reward of selecting the $i$th arm to be $r_i = (\mathbf{1}_{\text{succeed}} - h_i)^2$ and update $h_i$ to new value $\alpha \cdot h_i + (1 - \alpha) \cdot \mathbf{1}_{\text{succeed}}$ where $\alpha$ is the momentum of updating the history. Intuitively, this strategy assigns high rewards to tasks whose current outcome differs much from its history, and will assign zero reward to tasks which the RL agent always succeeds or fails on. We incorporate this momentum strategy in a bandit scenario algorithm [1]. Details in Supplementary.

## 3.3 Model

Our reinforcement learning framework is an modification of the AlphaZero setup. Specifically, a deep neural network $(\mathbf{p}, v) = f_\theta(s)$ with parameters $\theta$ takes a board state $s$ as input and predicts a vector of action probability $\mathbf{p}$ with components $\mathbf{p}_a = \Pr(a|s)$ for each valid action $a$ from the state $s$, and a scalar value $v$ indicating the estimated remaining steps to a goal state. In AlphaZero, $v$ is the winrate of the input game board. We adapt $v$ to represent the remaining steps to the goal. We take the set of all valid pushes as the action set $\mathcal{A}$ in our Sokoban experiments.

Given any board state $s$, a Monte Carlo tree search (MCTS) is performed to find the best action $a$ and move to the next board state. This procedure is repeated multiple times until either the goal state

is found or the length of current solution exceeds a preset limit. We set the length limit to 2000 to prevent infinite loops. At each board state $s_0$, we perform 1600 simulations to find the child node with maximum visit count. Each simulation starts from $s_0$ and chooses successive child nodes which maximize a utility function, $U(s,a) = Q(s,a) + \text{cput} \cdot \frac{\sqrt{1+\sum_b N(s,b)}}{1+N(s,a)} \cdot \mathbf{p}_a$, where $N(s,a)$ is the visit count of the action $a$ on the state $s$, $Q(s,a)$ is the mean action value averaged from previous simulations, and cput is a constant that controls the exploration/exploitation ratio. When a leaf node $l$ is encountered, we expand the node by computing all valid pushes from $l$ and creating new child nodes accordingly. Unlike traditional MCTS which uses multiple random rollouts to evaluate the expanded node, we use the $(\mathbf{p}, v) = f_\theta(l)$ from the neural network to be the estimated evaluation for the backpropagation phase of MCTS. The backpropagation phase updates the mean action value $Q(s,a)$ on the path from $l$ to $s_0$. Specifically, the $Q$ function value of nodes from $l$ to $s_0$ are updated with $v, v+1, v+2, ...$ accordingly.

After the RL agent successfully finds a solution for a instance, [22] suggested the information produced by MCTS on the solution path can provide new training data to further improve the policy/value network of the RL agent. Specifically, assume the found solution of length $n$ is $s_0, s_1, ..., s_n$ where $s_0$ is the input instance and $s_n$ is the goal state, the new action probability label $\hat{\mathbf{p}}$ for $s_i$ is the normalized visit count $N(s_i, a)$ for each $a$ in the action set $\mathcal{A}$, and the new value label $\hat{v}$ for $s_i$ is set to $n - i$ which reflects the remaining steps to the goal state.

In each iteration, the RL agent receives a batch of instances from the sample/bandit and attempt to solve them. For each successful attempt new training data is generated for the trainer to further update the parameters of the neural network $f_\theta$. Though we can generate training data from failed tries, we found it not impactful on the final performance. So that data is not used for training, which significantly reduces overall training time. The trainer keeps a pool of the latest 100000 improved policy/value data and trains 1000 minibatchs of size 64 in each iteration.

## 3.4 Curiosity-driven Reward and Graph Network

To further enhance the model, we augment MCTS with curiosity-driven reward and change the original convolutional architecture to the graph network. Though MCTS has a good exploration strategy, the curiosity reward can further help it avoid exploring similar states. We use random network distillation (RND) [3] as the intrinsic reward. When MCTS expands a new node, instead of setting the mean action value $Q$ to 0, we now set $Q$ to the intrinsic reward to encourage the model to explore nodes with high curiosity. The curiosity reward is especially helpful in multi-room cases without which the agent tends to push boxes around only in the initial room.

We use graph structure [5, 26, 25, 14] to extract features. Each board cell of the input cell is labeled by one-hot vector of seven categories: walls, empty squares, empty goal squares, boxes, boxes on goal square, player-reachable squares, player-reachable squares on goal square. Edges connect all adjacent nodes with linear mappings. See also Supplementary Materials.

## 3.5 Mixture of extra instances

We also test the benefit of adding extra Sokoban instances to the task pool of a hard instance. Specifically, after adding to the task pool sub-instances based on the input instance, we add extra *unrelated* Sokoban instances to the pool. Adding more data can make the "difficulty distribution" of all tasks smoother and the learning framework can solve harder instances with the knowledge learned from extra instances. In the experimental section we show how this strategy (with the "MIX" label) further boosts the performance of our automated curriculum approach.

## 4 Experiments

For our experiments, we collect all instances from the XSokoban test suite as well as large tests suited on [23] to form a dataset containing a total of 3,362 different instances, among which 225 instances are labeled with "Solved by none", meaning that they cannot be solved by any of four modern specialized Sokoban solvers. The solver time limit on the benchmark site is set to 10 minutes. However, because of the exponential nature of the search space, these instances generally remain unsolvable in any reasonable timeframe. We illustrate this further in subsection 4.2. Also, [11]

Table 1: The number of solved hard Sokoban instances (out of 225 unsolved hard instances) for different curriculum-driven RL strategies. The baseline (BL) model uses uniform subcase sampling with a convolutional network (CNN) to extract features. We add curiosity rewards (CR), a graph NN representation (GN), which replaces the default CNN in the BL, the Bandit (BD) subcase selection strategy, and, finally, combining all subcases of the 225 hard instances together (MIX) instead of solving each instance with its subcases separately.

|  | BL | BL+CR | BL+CR+GN | BL+CR+GN+BD | BL+CR+GN+BD+MIX |
|---|---|---|---|---|---|
| Solved | **30** | **52** | **72** | **115** | **146** |
|  |  |  | BL + CR | BL+CR+BD | BL+CR+DB+MIX |
| Solved | 30 |  | 52 | 103 | 105 |

Table 2: We first randomly shuffle the extra 3137 instances and fix the order. Starting with the best strategy in Table 1, we gradually add more of these instances to the curriculum pool.

| Added extra instances | 0 | 500 | 1000 | 1500 | 2000 | 2500 | 3137 |
|---|---|---|---|---|---|---|---|
| Solved hard instances | **146** | 148 | 156 | 160 | 165 | 173 | **179** |

considered the state-of-the-art Sokolution solver with a 12 hours time out on 6 of these problem instances, and was able to solve only 1.

## 4.1 Curriculum Learning

**Curriculum pool: unlabeled subcases** As was shown in [11], RL training using (unsolved/unlabeled) subcases with subsets of boxes can solve previously unsolved instances. They showed how their setup can solve 4 out of the 6 instances they considered after training on the subcases for up to 24 hrs per instance. In our experiments, we consider all 225 unsolved cases. For each instance, we generate 10,000 subcases by randomly selecting a number of boxes $k \leq N$ (where $N$ is the number of boxes in the original instance), and then randomly picking a subset of $k$ initial locations and $k$ goal locations. So, the instance to solve as well as its 10,000 subcases form the initial task pool for the curriculum strategy. The time limit for each instance is set to 12 hours on a 5-GPU machine and the whole learning procedure terminates once the original instance has been solved.

Our baseline (BL) setup is analogous to the [11] approach. In this setup, each batch of subcases in each training cycle is selected uniformly at random from the curriculum pool. This setting allows us to solve **30** out of the **225** hard cases (**13%**). Table 1 summarizes the results of our enhancements. We see how each enhancement, curiosity (CR), graph nets (GN), bandit (BD), and ensemble (MIX) boosts the performance of our approach, reaching **146** out of **225** hard cases (**65%**).

The bottom row of Table 3 considers the effect of the graph neural net representation. In particular, we see that using a mixture of training subcases (MIX) has little benefit when using the convolutional structure. In the single instance setting, all subcases share the same board layout (wall locations are unchanged). However, in the MIX setting, boards with different sizes and structures come in. Graph networks appear better suited to handle variable sizes and layouts and can therefore benefit from the MIX training pool, in contrast to the convolutional representation.

**Curriculum pool: adding unlabeled example instances** We further tested the benefit of adding the remaining extra 3,137 instances that can be solved by at least one modern solver. Unlike subcases, these instances do not necessarily share common board structure of the hard instances we are interested in. In a sense, these are "unrelated" practice problems or "exercises". (We don't use the instance solutions.) Table 2 shows the new added instances can yet further enhance the curriculum strategy, enabling the agent to solve **179** out of **225** hard instances (**80%**). As more extra instances are added, the percentage of solved instances increases which demonstrates the benefit of a large training dataset of "exercises," provided our bandit driven training task selection approach is used.

In summary, our series of enhancements, including the bandit-driven curriculum strategy, led us from **13%** of hard instances solved to **80%** solved. Moreover, instead of an instance-specific policy network, we obtain a single policy network that combined with MCTS can solve a diverse set of hard instances, which suggests the network encodes valuable general Sokoban knowledge.

Table 3: Running time of FF (general AI planner), **Fast Downward Stone Soup (winner of Satisfy-ing track 2018 International Planning Competition; a top performing general AI planner)** and Sokolution (top Sokoban specialized solver) on the instance Sasquatch7_40 with a **210**-step solution. Instances are built pulling backward from the goal and show increasing difficulty.

| FF (AI planner) | | | | | |
|---|---|---|---|---|---|
| steps | <40 | 50 | 60 | 70 | 80 |
| time | <10s | 3min | 21min | 2h | >12h |
| Fast Downward 2018 (AI Planner) | | | | | |
| steps | <60 | 70 | 80 | 90 | 100 | 110 |
| time | <21s | 5min | 17min | 58min | 3h | >12h |
| Sokolution (specialized Sokoban solver) | | | | | |
| steps | <110 | 120 | 130 | 140 | 150 | 160 |
| time | <20s | 52s | 3min | 22min | 4h | >12h |

## 4.2 Complexity of Search Space Analysis

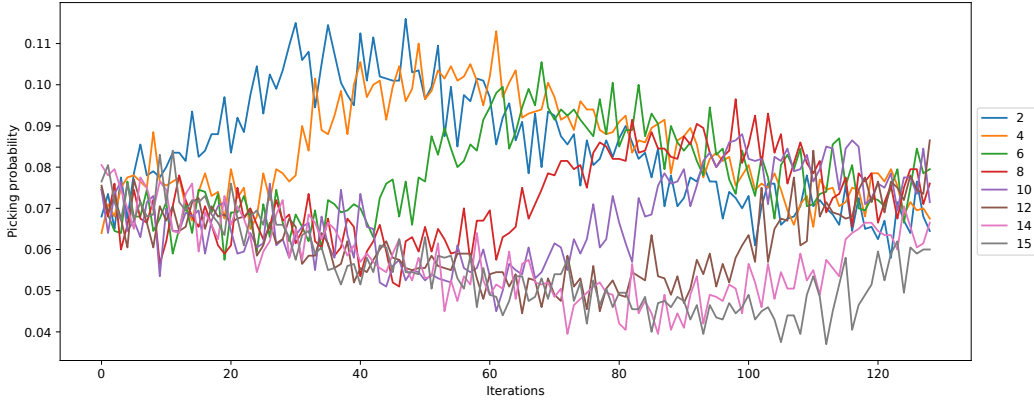

Figure 3: The probability of selecting $m$-box subcases by the bandit for $m \in [2, 15]$ on instance XSokoban_29 with 16 boxes. The bandit initially picks subcases with fewer boxes more frequently and gradually shifts to subcases with more boxes. Nevertheless, even at later stages, the bandit still mixes up subcases with different number of boxes, which is quite different from static strategies.

To get a better sense of the difficulty of the hard planning instances, we consider the scaling of other solvers. We randomly selected one of our solved instances, Sasquatch7_40, for which we found a solution with 210 steps. Since the problem is out of reach of other solvers, we considered the 210-step plan we found, and moved backwards from the goal state. Starting 10 steps back, we gradually increase the number of steps from the goal state until it takes more than 12 hours for search-based solvers to find any solution. Table 3 reveals the exponential scaling of FF [15], Fast Downward 2018 [21] and Sokolution. Uncovering the full 210 steps is out reach for both solvers and we estimate that Sokolution would take over 2 years (682 days) and FF would take over $5.7 \times 10^{10}$ years to solve the original instance. The truly exponential nature of the Sokoban plan search tasks is also clear from the scaling of the AI planners: roughly 20 years of AI planner development, from FF to Downward Stone Soup, gives us fewer than 30 extra steps in the plan length (2 hour time limit).

## 4.3 Evolution of Bandit Selection

Our experiments show that the bandit (BD) selection strategy solves more hard instances than uniform random selection. To further demonstrate the benefit of bandit, we visualize how the probability of picking each task evolves. We consider instance XSokoban_29 which contains 16 boxes, and for each $m \in [2, 15]$ we build 800 $m$-box subcases to form a task pool of 11200 instances in total. Figure 3 plots the percentage of several $m$-box subcases in each batch picked by the bandit. At the start, all subcases are sampled equally likely but then the probabiliy of selecting smaller box subcases (2-box, etc.) rises, because some of these are solvable early on and can provide good training signal and

the difficulty quantum momentum bandit strategy focuses in on those. The probability of sampling larger subcases (e.g., 14- and 15-box) goes down. In the later learning stages, the 14-box and 15-box selection probability rises, because some of those become solvable using what has been learned so far. Interestingly, small box subcases stay involved even at later training stages. This subcase mixture is an important factor for a smooth and fast learning process.

## 5   Conclusions

We provide a novel *automated curriculum approach* for training a deep RL model for solving hard AI planning instances. We use Sokoban, one of the hardest AI planning domains as our example domain. In our framework, the system learns solely from a large number of unlabeled (unsolved) subcases of hard instances. The system starts learning from sufficiently easy subcases and slowly increases the subcase difficulty over time. This is achieved with an automated curriculum that is driven by our proposed *difficulty quantum momentum* multi-armed bandit strategy that smoothly increases the hardness of the instances for the training of the policy deep neural network. We show how we can further boost performance by adding a mixture of other unsolved instances (and their subcases) to the training pool. We also showed the power of using a graph neural net representation coupled with a curiosity-driven search strategy. As a result, our approach solves 179 out of 225 hard instances (80%), while the baseline, which captures the previous state of the art, solves 31 instances (13%). Overall this work demonstrates that curriculum-driven deep RL — working without any labeled data — holds clear promise for solving hard combinatorial AI planning tasks that are far beyond the reach of other methods.

# 6 Broader Impact

We introduced a new framework for AI planning, which concerns generating (potentially long) action sequences that lead from an initial state to a goal state. In terms of real-world applications, AI planning has the potential for use as a component of autonomous systems that use planning as part of their decision making framework. Therefore we feel this work can *broaden the scope of AI,* beyond the more standard machine learning applications, for areas of sequential decision making. Our study here was done on a purely formal domain, Sokoban. In terms of *ethical considerations*, while this domain in itself does not raise issues of human bias or fairness, future real-world applications of AI planning (e.g., in self-driving cars and program synthesis) need to pay careful attention to the value-alignment problem [20]. To obtain *human value alignment*, AI system designers need to ensure that the specified goals of the system and the potential action sequences leading to those goals align with human values, including considerations of potential fairness and bias issues. In terms of *interpretability*, our approach falls within the realm of interpretable AI. Since although we use deep RL in the system's search for plans, the final outcomes are concrete action sequences that can be inspected and simulated. So, the synthesized plans, in principle, can be evaluated for human value alignment and AI safety risks. One final important broader societal impact component of our work is an interesting connection to *human learning and education*. We showed how our dynamic curriculum learning strategy, leads to faster learning for our deep RL AI planning agent. We showed how the curriculum balances a mix of tasks of varying difficulty and drives the learning process by staying on the "edge of solvability." It would be interesting to see whether such a type of curriculum can also enhance human education and tutoring systems. Our difficulty quantum momentum bandit driven strategy considers feedback from the planning ability of the system; similar feedback could be obtained from a human learner during the learning process. Finally, we were excited to see the benefit to the learning process of adding unsolved planning problems as "exercises." A further study of what starting pool of exercises is most effective may provide useful new insights for designing learning curricula for human education.

## Acknowledgements

We thank the reviewers for valuable feedback. This research was supported by NSF awards CCF-1522054 (Expeditions in computing), AFOSR Multidisciplinary University Research Initiatives (MURI) Program FA9550-18-1-0136, AFOSR FA9550-17-1-0292, AFOSR 87727, ARO award W911NF-17-1-0187 for our compute cluster, and an Open Philanthropy award to the Center for Human-Compatible AI.

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
