[Supplementary Material]

# A Novel Automated Curriculum Strategy to Solve Hard AI Planning Instances

## 1 Automated Curriculum Framework

Herein we provide additional details concerning the different algorithms comprising our *automated curriculum framework*. For the sake of completeness, the overview of the workflow of our ***automated curriculum framework*** is depicted in Figure 1 and the formal algorithm description can be found in Algorithm 1. The different algorithms include pointers to their sub-routine algorithms.

Figure 1: **The workflow of our automated curriculum framework, which is formally described in Algorithm 1.**

---

**Algorithm 1:** Automated Curriculum Learning framework overview

---

**Input:** Sokoban instance $\mathcal{I}$, solution length limit $L$, number of iterations $T$;
Call TaskPool($\mathcal{I}$) (Algorithm 2) to create a pool of sub-instances;
**for** $t = 1, ..., T$ **do**
    Use uniformly sampling (baseline) or difficulty quantum momentum bandit (Algorithm 3) to
    select a batch $B$ from the task pool;
    **for** $s \in B$ **do**
        **for** $i = 1, ..., L$ **do**
            Use MCTS($s$) (Algorithm 4) to select the best move $a$ for $s$;
            $s = $ next_state$(s, a)$;
            **if** *s is a goal state* **then**
                Generate data for training the policy/value network of the RL agent (i.e., for each
                  MCTS node (board) along the branch from the input state (root) to the goal state,
                  estimated distance to goal and distribution of visits to child nodes);
                Send "success" feedback to the sampler/bandit;
                Break;
            **end**
        **end**
        **if** *Solution not found* **then**
            Send "failure" feedback to the sampler/bandit;
        **end**
    **end**
    Train the policy deep neural network of the RL agent;
**end**

---

The master algorithm (Algorithm 1), starts by creating a task pool of sub-instances from the target instance (Algorithm 2) that we need to solve, and potentially from other unsolved instances to further boost performance (option MIX). Typically the task pool contains $100,000$ tasks or sub-instances. In each iteration, the sampler/bandit picks a batch of task sub-instances from the pool and passes it to the RL agent. A batch has typically 500 tasks or sub-instances (Algorithm 3). The RL agent, which is based on Monte-Carlo tree search (Algorithm 4), augmented with neural networks (CNN or GNN), attempts to solve these instances. For each instance in the batch, MCTS will seek a solution with a given resource budget, and for each successful solution generated, MCTS will also generate a chain of new training data for the policy/value deep network (trainer) to further update its network parameters. The MCTS success/failure status of each instance is sent back to the sampler/bandit to adjust its weights. Each successful attempt not only generates a valid solution but also improves policy/value data for the trainer to train the deep network of the agent. The trainer keeps a pool of size 100000 to store the most recent training data generated by MCTS, and train the network. Each training batch is uniformly randomly sampled. All experiments are done on a machine with 2x18 core Xeon Skylake 6154 CPUs and 5 Nvidia Tesla V100 16GB GPUs, and all training component use Adam with learning rate 0.002 as the default optimizer. The number of MCTS simulation $R$ is set to 1600 and the batch size $M$ that Exp3 samples in each iteration is set to 500.

---

**Algorithm 2: function** TaskPool($s$)

---

**Input:** Sokoban instance $\mathcal{I}$;
**Parameters:** The pool size $P$;
$p = \{\}$;
*boxes* = all initial box locations in $\mathcal{I}$;
*goals* = all initial goal locations in $\mathcal{I}$;
$N$ = size_of(boxes);
**for** $i = 1, ..., P$ **do**
    $n$ = UniformRand($[1, N]$);
    *rand_boxes* = A random subset of *boxes* with size $n$;
    *rand_goals* = A random subset of *goals* with size $n$;
    $p = p \bigcup$ SokobanInstance(*rand_boxes*, *rand_goals*) (i.e., first build a empty Sokoban
      instance with wall and player location of $\mathcal{I}$ unchanged, and then add *rand_boxes* and
      *rand_goals* to the board);
**end**
**return** $p$;

---

---

**Algorithm 3:** Exp3 to sample batches of instances using difficulty quantum momentum heuristic

---

**Input:** a task pool P;
**Parameters:** batch size $M$, exploration ratio $\gamma \in [0, 1]$, momentum $\alpha \in [0, 1]$, total iteration $T$;
**Initialization:** $N$ = size_of($P$), $\omega_i(1) = 1, h_i(1) = 0$ for $i = 1, ..., N$;
**for** $t = 1, ..., T$ **do**
    $p_i(t) = (1 - \gamma)\frac{\omega_i(t)}{\sum_{k=1}^{N} \omega_k(t)} + \frac{\gamma}{N}$ for $i = 1, ..., N$;
    Sample a non-repeated batch $B_1(t), ..., B_M(t)$ from $P$ according to the probability $\mathbf{p}(t)$;
    Run the RL agent on the batch $B(t)$ and train on the collected data;
    $r_j(t) = (h_{B_j(t)}(t) - \mathbf{1}_{\text{succeed on } B_j(t)})^2$ for $j = 1, ..., M$;
    **for** $j = 1, .., M$ **do**
        $\theta_j(t) = r_j(t)/p_{B_j(t)}(t)$;
        $\omega_{B_j(t)}(t+1) = \omega_{B_j(t)}(t) \cdot \exp(\gamma \cdot \theta_j(t)/N)$;
        $h_{B_j(t)}(t+1) = \alpha \cdot h_{B_j(t)}(t) + (1 - \alpha) \cdot r_j(t)$;
    **end**
**end**

---

Figure 2: A whole simulation of MCTS. White and red circles correspond to the Monte Carlo tree before simulation. A simulation starts from the root node and goes down until it reaches a leaf node (the lowest red circle). Then an Expand procedure follows and adds new child nodes (blue) beneath the expanded node.

---

**Algorithm 4: function** MCTS($s_0$)

---

**Parameters:** maximum solution length $L$, action set $\mathcal{A}$, number of MCTS simulations $R$, visit count $N(s, a)$;
**Input:** board state $s_0$ to seek a solution;
**for** $l = 1, ..., L$ **do**
    **while** $\sum_{a \in \mathcal{A}} N(s_{l-1}, a) < R$ **do**
        Simulate($s_{l-1}$) (Algorithm 5, see Figure 2 for the demonstration of a single simulation);
    **end**
    best_action = $\text{argmax}_a N(s_{l-1}, a)$;
    $s_l$ = NextState($s_{l-1}$, best_action);
    **if** $s_l$ *is a goal state* **then**
        **for** $i = 0, ..., l - 1$ **do**
            Add $\langle s_i, \text{Normalized}(N(s_i)), l - i \rangle$ to the trainer;
        **end**
        **Break**;
    **end**
**end**

---

## 2   Network Architecture

We use convolution neural network (CNN) as the baseline and compare its performance with graph network (GN) to show how different architecture setting affects the final result. The input to the CNN network is a $7 \times H \times W$ image stack consisting of 7 features planes with height $H$ and width $W$. Each feature plan corresponds to walls, empty squares, empty goal squares, boxes, boxes on goal square, player-reachable squares, player-reachable squares on goal square, respectively. We use the standard ResNet-18 to extract the feature of the input Sokoban instance. For graph network, we build the graph by assigning a node to each cell of the board input and adding edges to each pair of adjacent cells. Horizontal and vertical edges have two different labels to further enhance spatial information to GN. Each input board cell lies in seven different categories as the same in the CNN architecture. We learn a embedding from these seven categories to a feature vector of length 128 as the starting of the GraphNet (Algorithm 7). We set the number of iterations $D$ of graph network to 10. The output feature of graph network is further sent to two different multiple perceptions (MLP) to predict action probability ($\mathbf{p}$) and remaining distance $v$ of the input state $s$.

The output of CNN and GN consists of two predictions: action probability $\mathbf{p}$ and estimated remaining step $v$. We use *Softmax* activation for the probability $\mathbf{p}$. Since we set a maximum solution length $L$ throughout the experiment and $v \in [0, L]$, we normalize the step prediction to $[0, 1]$ and use *Tanh* activation for the value $v$.

---

**Algorithm 5: function** Simulate($s$) (See also see Figure 2).

---

**Parameters:** visit count $N(s,a)$, mean action value $Q(s,a)$;
**while** *$s$ not a goal state* **do**
    **if** *$s$ is a leaf node* **then**
        | Expand($s$) (Algorithm 3);
    **end**
    **else**
        best_action = $\text{argmax}_a Q(s,a) + \text{cput} \cdot \frac{\sqrt{1+\sum_b N(s,b)}}{1+N(s,a)} \cdot \mathbf{p}_a$;
        $s$ = NextState($s$, best_action);
    **end**
**end**

---

---

**Algorithm 6: function** Expand($s$)

---

**Parameters:** visit count $N(s,a)$, mean action value $Q(s,a)$;
$\mathbf{p}, v = f_\theta(s)$;
**for** $a \in \mathcal{A}$ **do**
    $N(s,a) = 0$;
    $Q(s,a)$ = CuriosityReward(NextState($s,a$));
**end**
**while** *$v$ not root* **do**
    $r$ = Parent($v$);
    $a$ = PreviousAction($v$);
    $Q(r,a) = (Q(r,a) \cdot N(r,a) + v)/(N(r,a)+1)$;
    $N(r,a) = N(r,a)+1$;
    $v = r$;
**end**

---

## 3   Curiosity Reward

We use random network distillation (RND) as our curiosity reward generator. Specifically, we build a graph network $f_\delta$ with randomized parameters and fix the parameters throughout the whole procedure. We then try to learn another graph network $f_\tau$ with different randomized initialization and try to make the prediction of $f_\tau$ as similar as the one of $f_\delta$. For each state $s$ that is requested for a curiosity reward, we set CuriosityReward($s$) = $l_2$ distance between $f_\tau(s)$ and $f_\delta(s)$. After each reward prediction, the input state $s$ is sent to a training pool of size 100000. At the end of each iteration, we train $f\_\tau$ for 100 epochs of batch size 64 to make its output closer to that of $f_\delta$ using squared error loss between the outputs of the two networks.

**Algorithm 7:** Graph Neural Network extracting feature from a Sokoban board

**Input:** graph $\mathcal{G}(\mathcal{V}, \mathcal{E})$, input features $\{x_v, \forall v \in \mathcal{V}\}$, depth $D$, neighborhood function $\mathcal{N} : v \to 2^{\mathcal{V}}$;

**Output:** A global feature of the graph $h_v^0 = x_v \forall, v \in \mathcal{V}$;

**for** $d = 1, ..., D$ **do**

    $g_v^d = \text{Aggregate}_k(h_v^{d-1}, \forall u \in \mathcal{N}(v))$;

    $h_v^d = \text{Normalized}(\text{ReLU}(W^d \cdot \{h_v^{d-1}, g_v^d\}))$;

**end**

**return** $\text{Average}(h_v^D)$ for $v \in \mathcal{V}$;