[Reviews · NeurIPS 2020]

Review 1

Summary and Contributions: The authors introduce a mechanism for automatically selecting subsets of problems to be used as a curriculum based approach for an RL agent. The ultimate aim is to solve difficult planning-like problems, and through a chain of bandit-based problem selection with MCTS policy training, they are able to solve many extremely difficult Sokoban instances. -{ Post Rebuttal Response }- I believe that the authors have fully understood the overstep in claims, and will update the paper to reflect this -- I'm subsequently raising my score to an accept. The rebuttal notes that gradually relaxing goal fluents as a way to generalize. I am almost certain that you'll find this to be largely inadequate, as many domains have few goal fluents and they just don't provide a natural scaling. It is by no means trivial to generalize to the full space of domains, and I wish the authors the best of luck in trying to do so! The rebuttal has a comment "GET NUMBERS!!". I assume this is a note to yourselves left in error, but I would echo it back as a general suggestion for the appendix if accepted. It's fairly easy for you to establish dominance (across planning, RL, and domain-specific solvers). With that in hand, picking apart how the approach is behaving is of great interest. If you are to include Table 1, I would suggest making it wider, unifying the column headings, and adding "-" for time-outs. As it currently stands, it's easy to misread. Finally, you do not need to throw out the FF data. Include it as an indication of how the field has progressed over (~)20 years. Then Sokolution is offered as a domain-specific implementation that was the previous best, and quite expectedly outperforms domain-independent approaches. Then your approach, which has a general framework but domain-specific abstractions for training, clearly makes a leap beyond that. One major take-away readers will want is how to apply this in their own domain. What are the properties of the abstractions you defined for Sokoban? Is there something special about the diversity of the problem instances or the gradual increase of difficulty? These are the insights that should follow from the "GET NUMBERS!!" comment above that would make this a very strong piece of work.

Strengths: The obvious core strength of this paper is it's formidable results in the Sokoban domain. It is certainly a hard setting for many techniques, and the number of previously unsolved instances that now have become solved clearly indicates a strength with this work. Another strength, which I believe is mistated on lin 46 and under-emphasized more generally, is that the approach doesn't require the difficulty of the sub-instances to be labeled (line 46 contrasts with [11] by saying that this work /does/ require the labeling, but I think that is reversed). This is particularly appealing when there is no real way to assess the difficulty of a problem, but an easy mechanism to generate them.

Weaknesses: The biggest issue I have with the paper is that it is very domain specific, while the claims do not make this clear. It is entirely constructed to solve sokoban problems, from the encoded representation of the game, architecture(s) used for the encoding, subproblem creation, simplification on the reachable state space (ignoring moves), hand tailored features for the curiosity and GNN components, etc. This is all fine if the ultimate goal is to solve sokoban problems that no approach has been able to before, but claims to the general nature of the work are misleading. For the approach to be truly a general one, there must be (1) a domain-agnostic representation (you can steal the planning one for this -- PDDL); (2) a domain-agnostic way for generating subproblems of varying difficulty (the trick used for sokoban is very sokoban specific); and (3) and empirical demonstration on a number of domain settings to demonstrate what it is that you're proposing. I realize some of the criticism apply equally well to published work (i.e., [11]), but it doesn't lessen the issues with this work. One such example is the re-writing of the domain to abstract away movement. Yes, it is a trivial procedure to compute the fully reachable set of cells using "move" actions, and thus reducing the problem to "push" actions, but this is extremely domain-specific. In the context of solving sokoban problems that no one else ever could, that's ok (every trick is fair game), but it makes the comparisons to domain-independent approaches largely meaningless. As far as I'm aware, there is no encoding of sokoban in PDDL that abstracts away the movement actions without resorting to derived predicates to compute reachability (and even then, you're still relying on a different formalism). This also brings up the issue with using FF as a comparison. As pointed out by the authors, there have been regular contests held in the field of automated planning, and the FF system has been surpassed by many others over the last 2 decades and roughly half-a-dozen contests since. The assessment at the end of 4.2 seems wildly arbitrary, and speaks to only one planning system (known to suffer in it's primary phase on the sokoban benchmark). Whatever analysis you conduct of this form, no matter how well founded, will be completely irrelevant for the dozens of other planners you don't consider. Given how strong the results actually are in this paper, I think the current rhetoric and comparison only hurts the argument unnecessarily. A simple fix would be to use state-of-the-art planners (see the most recent IPC results), provide them a full 12hr limit (chance are the memory will be violated first), and lean on the fact that domain-independent planning is typically worse performing than domain-specific solvers -- and thus, expectedly worse than what you've introduced.

Correctness: The claim of "subcase mixture is an important factor for a smooth and fast learning process" (in reference to Fig3) is not substantiated. All the data indicates is that the curriculum selection strategy leads to this subcase mixture, and it fails to actually support your claim. There is another claim on line 109 that "the main focus of AI planning is on solving previously unsolved problems in reasonable time limit instead of accelerating solving easy instances". This is very much not the case -- particularly with planning-in-the-loop systems being a main source of planner usage, I would venture to say that the claim actually has things reversed. The planning techniques that would be used for solving a problem at any cost are very different than the default ones you will find, for precisely the reason why this claim doesn't hold: planners are designed to solve a certain level of difficulty as quickly as possible with good quality solutions. Here, quickly usually refers to fractions of a second to an hour at most. I believe there is an error in the definition of the curriculum learning setup -- the update rule should take the reward into account rather than the success on the current instance.

Clarity: It wasn't immediately clear how the bandit approach was coping with the naturally shifting distribution of the problems it is selecting from. This only became clear after going through the supplementary, but ultimately there is far too little information provided in the paper on this aspect of the work. Both from a rhetorical point of view, and an evaluation point of view. Generally, I think there is not nearly enough detail about what is unique in this work compared to [11]. I have to admit that I was unaware of [11] (it's only scheduled to appear at a conference in the new year), but looking through that work demonstrates how many things are similar. The elements that are unique to this paper should be elaborated on, and this can be done at the expense of the elements common to [11].

Relation to Prior Work: The relation to [11] is stronger than one would guess by reading the paper. The setup is extremely similar, and the core difference claimed in curriculum strategy isn't experimentally evaluated. As far as I can see, the BL selects problems uniformly. This is an ok baseline, but it fails to capture the nature of increased difficulty from [11]. Figure 3 only demonstrates the nature of how the selection strategy evolves; demonstrating that it is seemingly different than that of [11]. It doesn't establish that the new selection strategy is better. Also, claims that the strategy from [11] is "hand tailored" seem a bit extreme -- it is a general approach that gradually increases the difficulty. It feels a bit odd that the authors didn't try to employ an ExIt / AlphaZero style of approach with the setup being so close to it already. There is an argument given the paper that those approaches require the signal from game wins in a 2-player setting, but the broader framework can certainly apply here: having an apprentice policy trained on an MCTS (using an older version of the policy), and then swapping the MCTS policy for the newly trained one to iterate. Finally, I am not entirely familiar with the curriculum based learning (both in the RL sense and in the education sense), but it strikes me as a natural strategy to select problems on the threshold of being solvable (or not) by the target learner. Given that this is a central aspect of this paper, some contrast there (or a claim to the novelty of it) seems warranted.

Reproducibility: Yes

Additional Feedback: My strongest suggestion would be to either (1) scale back the claims (in the title, abstract, and text) to match the content of the contribution -- something tailored for the setting of sokoban; or (2) expand on the work to fully explore the hypothesis surrounding the solvability threshold in curriculum learning and demonstrate this on a variety of planning domains. Aside from the questions raised in other parts of this review, there are two further aspects I was curious about: How would you expect this to work in out-of-distribution problems? There are many ways to pose a problem, and I can imagine, for example, only having training data that deals with a single room, and testing data that uses multiple rooms. Do you think that the subproblem creation scheme devised for sokoban naturally guards against this by providing a wide distribution in the curriculum? There is an underlying hypothesis that (1) the introduced sampling mechanism selects problems that are on edge of being solved by the MCTS; and (2) these are the best problems to select for a training regime. Have you explored this hypothesis directly? You can invest the resources to get a sample-based sense of what the true solvability likelihood would be for each problem, given a fixed policy, and then use that "oracle" information to explore both points above.


Review 2

Summary and Contributions: This paper proposes a curriculum learning method to select training instances for RL agent. The paper focuses on the Sokoban tasks. It further generates a number of sub-tasks from each task by sub-sample a subset of boxes/goals and form a pool of tasks. During each iteration, it samples a subset of tasks that are at the frontier of learning which are not too hard or too easy to the current agent by a bandit algorithm, and then train the agent to finish each selected task. The agent adopts a Monte-Carlo tree search strategy to find a successful trajectory and use it to update its policy/value networks. The paper also proposes to use graph neural nets to extract features, add curiosity reward to encourage exploration, and mix extra tasks together with the sub-tasks to form a more diverse pool of tasks. In experiments, they show the proposed bag of strategies can solve more Sokoban instances in shorter time.

Strengths: 1) It develops a bag of techniques for solving Sokoban tasks more effectively. 2) The idea of curriculum learning by selecting the samples at the learning frontier makes sense.

Weaknesses: 1) The major idea of selecting instances in this paper is not novel. It is natural to select tasks that are solvable but not too easy or too hard to the current agent during training. There exist several works in RL using the same idea, for example, Automatic Goal Generation for Reinforcement Learning Agents by Florensa et al. 2018. 2) The proposed bandits algorithm does not have any theoretical justification. The proposed reward is simply the deviation of the observed success comparing to the running mean of success over history, and the momentum parameter alpha controls the soft window width of the running mean. Is the performance sensitive to alpha? Does different instances need different alpha since the agent could make progress on them with different speeds. These questions are important to understanding how the selection criterion/reward helps to accelerate the learning. 3) The sparse reward problem is handled by MCTS instead of the proposed curriculum. But MCTS can be very expensive especially in early stages. 4) The empirical analysis of the experiments is insufficient. The main results are simply the final times and #solved instances. As a curriculum learning method for RL, it is necessary to have plots showing the improving curve for the success rate/average reward during training. In addition, it is important to have more visualizations and analysis of the selected tasks, graph neural nets learned features, and curiosity reward. 5) Basic comparison lacks in most results. There is only one baseline, which is FF, when reporting the running time. Except this, no baseline is presented and compared with. Hence, it is not convincing that the proposed method indeed outperforms most other possible RL methods on Sokoban tasks. 6) The proposed method (e.g., the generation of subtasks) is specifically designed for Sokoban and might be hard to be extended to other RL tasks.

Correctness: It is a pure empirical paper without rigorous problem formulation or theoretical justification. So it is hard to justify its correctness. The claims of motivations are correct. The empirical methodology is questionable since it does not provide enough details and comparisons.

Clarity: The paper is clear in its high-level ideas but the clarity can be significantly improved.

Relation to Prior Work: It covers several lines of related works but needs to add more related works on different strategies/criteria of instance/task selection, especially the ones using a similar criterion.

Reproducibility: Yes

Additional Feedback: ----After Rebuttal ------ Below is the major concern from my perspective. 1. The aim of the momentum (stated in line 164-169) is the same as the Goals of Intermediate Difficulty (GOID) in GoalGAN (Florensa et al. 2018). 2. The proposed momentum method was claimed as a Bandit algorithm without providing an analysis of its regret bound. 3. The additional experimental comparison in the rebuttal provides more support. For this reason, I would like to raise my rating from 4 to 5. However, reporting the learning curves are necessary for curriculum learning papers.


Review 3

Summary and Contributions: This paper proposes a new set of strategies for solving hard Sokoban instances (and potentially other hard planning problems) by training a deep RL agent using a curriculum of of simpler instances. The strategies include: 1. Use of random network distillation to provide auxiliary reward during training. 2. A bandit-based selector for instances that is rewarded for presenting the agent with instances that have "surprising" results (e.g. an instance on which the agent succeeds after having previously failed most of the time, or an instance on which it fails after consistently succeeding). 3. Use of a graph network with connectivity specifically tailored to the topology of Sokoban environments. 4. The idea of sampling instances for the curriculum generator from both synthetically generated permutations of a single original instance (produced by randomly removing blocks and goal positions from the instance), and from other instances in the dataset. Results on a large dataset of Sokoban problems demonstrate (1) superior coverage to what (AFAICT) was the previous state-of-the-art, and (2) that each proposed method substantially improves performance.

Strengths: Strengths: - The greatest strength of this paper is its strong empirical results on Sokoban, which is widely considered to be a challenging domain. Further, the ablations show that each individual component of the larger system (MIX, BD, CR, GN) adds some marginal increase to the overall performance, with the greatest increase coming from the bandit-based curriculum generator. - The bandit-based curriculum generation approach, which I view as the main contribution of this paper, does not appear to exploit any features specific to the Sokoban domain, and so is likely to generalise to other hard planning problems. The results (in particular Figure 3) support the claim that the bandit algorithm is able to generate an intuitively meaningful curriculum across the course of training. - The success of RND in this domain is surprising, given that IW(1) novelty search does not appear to work well (see Geffner & Geffner's "Width-Based Planning for General Video-Game Playing" in AIIDE 2015). The relative value of different novelty measures in classical planning would be interesting to investigate in future work (although it's possible that this particular result is just a quirk of Sokoban and/or the chosen RL algorithm—see below).

Weaknesses: Weaknesses: - It's unfortunate that Sokoban is the _only_ domain used for evaluation, and also unfortunate that there is only a single, domain-specific strategy used to generate training tasks (specifically, generating new Sokoban instances from predefined ones using a subset of boxes/locations). Thus it is possible that some of the proposed strategies do not work as well in other domains. For example, the interaction between GN and MIX in Table 1 is not something I would have predicted beforehand, and may be an artifact of the domain. Indeed, the motivation for MIX in Section 3.5 is relatively weak: this seems like a strategy that only makes sense with the particular baseline strategy that the paper used for generating a curriculum. The paper still makes a novel & interesting contribution, although perhaps are more accurate title would be "A Novel Automated Curriculum Strategy to Solve Hard Sokoban Instances". - There's no indication of inter-run variance in any of the results. Given how high the variance of RL algorithms can be, it would be good to include this in the camera-ready (or any future re-submissions), or at least provide some qualitative indication of how much results vary between runs. - The comparison in Section 4.2 is somewhat misleading: while it gives an accurate picture of the difficulty of solving these instances, it does not give an accurate impression of the capabilities of non-learning approaches. I'm not sure about Sokolution, but FF is definitely constrained to a single CPU core. In contrast, the evaluated method has access to five GPUs. Assuming 250W/GPU (GTX-1080Ti) and 12 hours of traiing, the proposed method gets 15kWh of compute, whereas if the baselines are run on one core of an eight-core CPU with 150W TDP (made-up numbers, but fairly typical), then they receive only 150/8*12=225Wh of compute, which is 1.5% of the learning solver. (I expect similar results with other metrics like hardware cost.) This disparity should be acknowledged somewhere in 4.2. - The only baseline is an analogue to the method of Feng et al. [11] (referred to as BL). While it seems unlikely that domain-specific heuristic search planners could be competitive in this setting, it would at least be nice to compare to Groshev et al.'s supervised leapfrogging strategy [13], and any other methods that belong in a similar class to this one.

Correctness: The claims and methods seem broadly correct, with the exception of concerns about the validity of inferences drawn in Section 4.2, the lack of inter-run variance measures, and the missing hyperparameters in the appendix (see weaknesses & clarity sections, respectively).

Clarity: Yes, the paper was straightforward to follow. Some minor quibbles: - Some typos: "tests suited on" -> "test suite from" (L233), "a instance" -> "an instance" (L200), one "DB" should be "BD" (Table 1), "table 3" should be "table 1" (L255). - Referring to past work with [numbers] makes it slightly harder to follow the discussion. In my opinion would be preferable to refer to papers by the name of the algorithm they present (if possible), or even by the names of the authors. - The "broader impacts" section is perhaps overly broad. For example, it claims that this systems is "interpretable" because it produces a plan, but it's hard to imagine a Sokoban solver (or, more generally, deterministic planner) that would _not_ have this property. The human learning point seemed more directly applicable to the specific contribution of this paper. - The experiments/appendix seem to be missing some of the hyperparameters used for evaluation, such as gamma and alpha for Exp3.

Relation to Prior Work: Yes, and I'm not aware of any closely related work which the authors missed or mischaracterised.

Reproducibility: No

Additional Feedback: Rated 7 because the proposed algorithm seems technically sound and achieves strong results. My main source of unease is the fact that Sokoban is the only evaluation domain. Things that would make me advocate more strongly for this paper after the author rebuttal: - Addressing (or committing to address) the issues in "clarity" (in particular the hyperparameters, assuming I haven't simply missed them—this is why I answered "N" for the repro question) - Explaining why the MIX/GN components are likely to work in other domains beyond Sokoban (if they are). - Providing (or committing to provide) results with some measure of inter-run variance. - Addressing the fairness issue in Section 4.2. ----------------- POST-REBUTTAL FEEDBACK: Thank you to the authors for committing to reduce the scope of the claims at the beginning of the paper to clarify that the main contribution is solving hard Sokoban problems. I still believe this is a good submission for the reasons outlined above. However, I would encourage the reviewers to address the outstanding issues above, particularly measures of inter-run variance and discussion of which components of the algorithm are likely to work in other domains (per R1's updated review).


Review 4

Summary and Contributions: I thanks the authors for the rebuttal. If the paper is accepted, it is essential that the authors integrate the feedback from the reviewers as much as possible. Among others, the authors agree to present the scope in the recommended way, as a Sokoban-specific paper, rather than general planning. The paper focuses on solving difficult combinatorial-search problems such as Sokoban instances. Building on top of recent work, the approach involves Monte-Carlo Tree Search (MCTS) combined with a deep learning neural architecture that provides probability scores for actions applicable in a state, as well as a value of a given state. Deep learning architectures considered include a convolutional neural net and a graph neural net. As one of the main novelties, the paper introduces a strategy to select the instances to use in training at a given point in time. The empirical evaluation provided shows that the approach presented solves more difficult instances from a standard dataset than state-of-the-art solvers.

Strengths: The performance appears to be strong, with a significant number of difficult instances being solved automatically for the first time, according to the paper. These are solved with a unique learned policy, as opposed to a different policy for each instance. The paper is clear and relatively easy to read.

Weaknesses: The title is misleading and so is the claim that this work is focused on AI planning. All this work is focused on one single domain. I understand that these ideas could also be applied to other planning problems, but until this is demonstrated with an implementation and experiments, the claim about addressing planning as a broad area is not justified. Tables and Figure 3 lack visual clarity, being difficult to grasp their main point just through a quick glance. This needs to be improved. The analysis in Section 4.2 needs to be broader than just one instance.

Correctness: Assuming that comments provided in this review are addressed (such as the one about planning vs Sokoban), the correctness looks okay.

Clarity: Yes. Please see a few minor issues in the comments to the authors.

Relation to Prior Work: The related work survey looks reasonable. Some older work on Sokoban includes: Andreas Junghanns, Jonathan Schaeffer: Sokoban: Enhancing general single-agent search methods using domain knowledge. Artif. Intell. 129(1-2): 219-251 (2001) Adi Botea, Martin Müller, Jonathan Schaeffer: Using Abstraction for Planning in Sokoban. Computers and Games 2002: 360-375

Reproducibility: Yes

Additional Feedback: Typos: Line 14 ("approach" mispelled) Line 180: "an modification" Line 200: "a instance" Using a reference instead of a name in a sentence is a bad style as far as I can tell. For example: "[10] first proposed", "[12] used a..." etc. The correct phrasing is: "Elman [10] proposed" etc.

[Author Response · NeurIPS 2020]

Table 1: New version of Table 3 in our submitted paper. Running time of FF (general AI planner), Fast Downward Stone Soup (winner of Satisfying track 2018 International Planning Competition; a top performing general AI planner) and Sokolution (top Sokoban specialized solver) on the instance Sasquatch7_40 with a 210-step solution. Instances are built pulling backward from the goal and show increasing difficulty.

| FF (AI planner) | | | | | |
|---|---|---|---|---|---|
| steps | <40 | 50 | 60 | 70 | 80 | |
| time | <10s | 3min | 21min | 2h | >12h | |
| Fast Downward 2018 (AI Planner) | | | | | |
| steps | <60 | 70 | 80 | 90 | 100 | 110 |
| time | <21s | 5min | 17min | 58min | 3h | >12h |
| Sokolution (specialized Sokoban solver) | | | | | |
| steps | <110 | 120 | 130 | 140 | 150 | 160 |
| time | <20s | 52s | 3min | 22min | 4h | >12h |

We thank the reviewers for their careful and detailed reviews. **I Scope of contribution** We agree with several of the reviewers that we stated the title and introduction too broadly about AI planning, while we focus on the Sokoban domain. Following the suggestion by reviewer # 3, we will change the title to "A Novel Automated Curriculum Strategy to Solve Hard Sokoban Instances." We did select this domain because we know this problem to be an extremely hard combinatorial AI planning task, with many open unsolved instances that beyond the reach of all other approaches (both specialized and general solvers). Our approach solves those instances. We will also narrow the scope of the introduction. We should note though that we believe the ideas we used are sufficiently general to extent to other planning domains. E.g., to solve a very hard unsolved planning instance, we can create a series of easier sub-instances by removing grounded predicates from the goal state. However, this is for future work.

**II Comparison to State-of-the-Art Solvers and Baseline** The reviewers are totally correct that we should have used a more recent AI planner. We ran experiments with the 2018 winner of the planning competition, Fast Downward Stone Soup. See results in the new table 3 above. We see indeed a significant improvement of about 30 more steps over 20 years of AI planning technology. Within 12 hrs compute time, FF cannot find plans further than 80 steps from the goal; Fast Downward cannot go further than 110 steps away; the specialized Sokolution solver cannot go further than 160 steps away. Our approach finds a solution from the original start state at 210 steps away. We again stress that we are solving instances that are not solved by any other method.

Reviewers # 1 and # 2 suggest we only compare to FF and ask about a baseline and other solvers. First we note that we can only compare to the "weakened" instances, with initial states placed closer to the goal, because our real contribution is in solving the full original instances that are not solved by any previous method, including the specialized solver Sokolution (which itself already greatly outperforms general AI planners or any other known RL results on Sokoban). Earlier work on RL for Sokoban, eg by the DeepMind group, could only solve some of the known instances that are trivial for eg Sokolution (solved in seconds). So, we do believe our approach is an advance even for RL.

**III Curriculum Strategy and contrast with paper [11]. GET NUMBERS!!** As various reviewers noted, a key novelty is the new curriculum strategy combined with sub-instance approach but also several other innovations as highlighted in the ablation studies. Overall, we solve 179 of the total of 225 known open problem instances. The approach presented in [11] only solves dozens of the open problems (in under 12 hrs each). So, we do believe our framework significantly extents that of [11]. More work can be done on studying the bandit instance selection but one core finding is that we do not have the "forgetting" problem as observed in [11]. Our RL policy continues to improve without "forgetting" how to solve earlier instances. This effect is due to our bandit strategy that keeps some easy instances around to retain the basic strategies.

As pointed out by reviewer # 1, we will state more clearly that we learn from unsolved (unlabeled) sub-instances. This is a core feature of the approach.

The work on automated goal generation in robotics (Florensa et al. 2018) is related (reviewer # 2). However, there the emphasis is on learning to operate in more diverse settings. Our approach with the curriculum training and pool of sub-instances is needed to build towards solving a particularly hard unsolved instance. The bandit strategy carefully pushes the training pool to increasingly challenging problems, to finally solve the original hard instance.

We thank Reviewer 3's four detailed and constructive questions. For the clarity, we have stated hyperparameters in the main paper in different places and we will put them together for more clarity. Though our method used more GPUs, the main motivation of Table 3 is to show the exponential scaling of these solvers as the size of sub-instance increases so that there is no hope for these solvers to solve the original instance due to exponential explosion.

[Meta-Review · NeurIPS 2020]

The paper presents a curriculum-based approach for training RL agents on difficult planning task, such as Sokoban. The paper contributes a collection of approaches working together (from sub-task sampling, to MCTS as means to collect training experience) to solve complex tasks. The paper's contributions are novel and empirical evaluation shows strong results. The work is well positioned within the related works. Per the reviewers' comments and the authors' responses, in the camera ready version, the authors should: - Reframe the scope of the paper to around solving difficult Sokoban problems instead of general planning, - Discuss generalization to the other domains.